# Effects of the Number of Crested Cushions in Runzhou White-Crested Ducks on Serum Biochemical Parameters

**DOI:** 10.3390/ani13030466

**Published:** 2023-01-28

**Authors:** Jiying Lou, Qixin Guo, Yong Jiang, Guohong Chen, Guobin Chang, Hao Bai

**Affiliations:** 1Joint International Research Laboratory of Agriculture and Agri-Product Safety, The Ministry of Education of China, Institutes of Agricultural Science and Technology Development, Yangzhou University, Yangzhou 225009, China; 2College of Animal Science and Technology, Yangzhou University, Yangzhou 225009, China

**Keywords:** Runzhou white-crested ducks, crested cushion, serum biochemical parameters, immune indicators

## Abstract

**Simple Summary:**

The Runzhou white-crested (RWC) duck is a unique breed with excellent ornamental and economic value. It has complex feather protrusions, collectively known as crest cushions, which mainly comprise prominent soft tissues covered by skin and feathers. In this study, the number of crest cushions and various serum biochemical parameters of RWC ducks were observed and measured. The results showed that the number of crest cushions could be used to directly reflect the size of the crest. As the number of crest cushions increased, the weight and mortality increased. In addition, the number of crest cushions had a positive effect on various serum parameters, such as various mineral elements as well as triglyceride and immunoglobulin levels. Therefore, it is preliminarily speculated that the number of crest cushions may indirectly affect the health and immune performance of RWC ducks.

**Abstract:**

We investigated the effects of crest cushions in Runzhou white-crested (RWC) ducks. A total of 322 duck eggs were collected for incubation; 286 eggs were fertilized, and 235 RCW ducks were hatched. All the RWC ducks were weighed after 100 days and counted, and the volume of the crest cushion was measured. The number of crest cushions was positively correlated with the body weight, volume of the crest cushion, and distance from the mouth (*p* < 0.05). The serum Ca, Mg, Fe, Cu, Zn, and Se contents in the multiple-crest-cushion group were significantly higher (*p* < 0.05), as were the levels of triglycerides, immunoglobulin A, immunoglobulin G, immunoglobulin M, and immunoglobulin D (*p* < 0.01). The opposite results were seen for glycosylated low-density lipoprotein (*p* < 0.01). Propionic acid and acetic acid contents differed significantly between the two groups (*p* < 0.05), as did butyric acid content (*p* < 0.01), being higher in the multiple-crest-cushion group. Thus, an increase in the number of crest cushions coincided with a change in various serum biochemical indicators. The number of crest cushions might be involved in regulating various mechanisms of RWC ducks and might have an immunoregulatory effect.

## 1. Introduction

The crested-cushion white duck is a unique breed with complex plumage protrusion traits collectively known as the crown. It has been documented in various historical books; however, this high-quality variety was gradually lost over time and did not reappear until 2005, when it was discovered by Yangzhou University and Zhenjiang Tiancheng Agricultural Technology Co., Ltd. After rescue collection and collation, it was successfully restored in 2013 as a Runzhou white-crested (RWC) duck population, which was consistent with the records of historical books examined by the China Livestock and Poultry Resources Committee as a unique local breed in Jiangsu Province. The breed is characterized by a cluster of clumped feathers protruding behind the head and shaped like a fluffy ball. There is no significant difference in appearance between males and females, both having moderate body size, symmetrical structure, and compact and narrow bodies. The feathers are white in color and tight.

Avian breeding accounts for a large proportion of the total agricultural output value in the country and worldwide. Meat duck breeding has gradually occupied a major position in the waterfowl industry in China, and annual production continues to rise [1]. High-quality ducks play a crucial role in the Chinese poultry market and are an important field of interest for research. RWC ducks are ornamental duck breeds with high meat, egg, and medicinal value. Because crest cushion ducks account for a low proportion of the population at the beginning of breeding, the numbers and sizes of crest cushions in different ducks are different during population expansion.

The feather crest cushion of domestic ducks reflects a head integument that inserts into the subcutaneous thickened skin of the parietooccipital region [2,3]. In poultry, chickens, ducks, pigeons, parrots, and some wild birds have been found to have crest cushion traits; however, some studies have found that duck crested cushions are different from those of chickens, and there are morphological differences between them [4,5,6,7]. Wang et al. [8] found that chicken crested-cushion formation is due to vesicular expansion of the cranium, resulting in abnormal growth of the telencephalon under its cranium, forming an hourglass-shaped brain hernia structure. Frahmm et al. [9] suggested that this abnormal increase in brain tissue in crested-cushion chickens might be related to their similarly large ventricles. Our group previously found that there are fat-body-like lipomas in the brains of crested-cushion ducks, with some cranium missing or incompletely closed in the head and a small part of fat body tissue and brain tissue spillage, which is consistent with the findings of Bartels et al. [2,10,11].

Vertebrate serum contains many enzymes, nutrients, ions, hormones, and waste products, which are biomaterials that are widely used to determine animal health status [12]. Other serum biomarkers similarly indicating animal health status are glucose, triglycerides, cholesterol, and proteins. Short-chain fatty acids (SCFAs), mainly comprising acetic acid, propionic acid, and butyric acid, are important metabolites of the gut microbiome and play a crucial role in the normal life activities. They are the energy source of some cells, affect the blood lipid and blood glucose levels, and can regulate the colonic environment, inflammation, metabolism, immune function, and tumor cell proliferation and apoptosis [13,14,15,16]. 

So far, most studies have detected the effects of different feeding methods and different feed additions on serum biochemical parameters of animals, and there is still a great gap in the study of the effects of different traits [17,18]. The duck is a carrier and donor of various mineral elements, and the deposition and metabolism of mineral elements are closely related to the growth, meat quality, feed conversion rate, and disease resistance of meat ducks. Therefore, it is important to study the deposition of mineral elements in RWC ducks to promote their production and improve meat quality. As to our observations, the hatching success rate of RWC ducks during the population expansion was lower than that of common ducks. We initially speculated that this is related to the specific crest cushion trait and further investigated the effect of crest cushions on RWC ducks to select better quality RWC ducks in the future. In this study, we selected important immune indicators, mineral element parameters, and SCFAs for the analysis of serum biochemical indicators to determine whether the number of crested cushions affects various serum biochemical indicators related to RWC duck health.

## 2. Materials and Methods

### 2.1. Ethics Statement

All experimental ducks were approved by the Yangzhou University Animal Protection Association (approval number: 151-2014). All experimental ducks were managed and handled according to the guidelines established and approved by the Animal Care and Use Committee of Yangzhou University. All possible efforts were made to reduce the suffering of animals.

### 2.2. Animals and Experimental Design

The RWC ducks used in this experiment were obtained from TianCheng Agricultural Technology Co., Ltd. We collected 322 RWC duck eggs for hatching. All breeding male and female ducks were healthy individuals and the ducks we raised had been injected with the vaccine; in addition to avoiding excessive egg contact with external pathogens, we collected embryos at the first time after laying eggs in female ducks, comprehensively disinfected the incubation box before hatching, and at the same time, the incubation box also ensured normal work. In order to eliminate the pathogen from interfering with the embryo during the whole hatching process, we also strictly followed the production requirements for disinfection. The eggs were candled on the eighth day after collection and the unfertilized and dead eggs were removed. Ducklings successively hatched on approximately day 28 of incubation. The duck shed was fumigated with formalin and potassium permanganate 1 week before housing and ventilation was restored after 24 h of closure. Ducklings were kept at 35 °C in the incubator before entering the cage, humidity was controlled between 70% and 75%, and all ducks were housed on ground covered with dry rice husk bedding; room temperature was maintained at 33–35 °C for 1 week during the brood period, then decreased by 2–3 °C per week, and out of temperature control after 3 weeks, during which stocking density remained at 15 ducks/m^2^. Stocking density gradually decreased with increasing age until 42 d of age, when they entered the finishing period and remained at 5 animals/m^2^. In the duck house, they had the same living environment, with 24 h of uninterrupted light (natural light during the day and incandescent light at night) and equipped with automatic drinkers and feeders so each duck had free access to food and water. The ducks were reared on the same diet; feed nutritional value requirements are shown in Table 1, and feed was purchased from Changzhou Dajiang Feed Co., Ltd. Depending on the weather and status of the duck flock, ducks were allowed out of the duck house for free range activities. 

After 100 days of feeding, the feed was removed in the evening, and the number of crested cushions from 235 hatched RWC ducks were counted. One crested cushion was recorded as a single-crest group, and two or three crested cushions were recorded as multiple-crest groups. All individuals were weighed and recorded for statistical analyses. 

### 2.3. Crested Cushion Data Collection

The eggs that did not successfully hatch were collected separately and broken, and the number of embryos inside and the number of crested cushions were counted. Unhatched embryos were divided into a single-crest group and multiple-crest groups. Based on this, ducks were weighed using a weighing scale. A Vernier caliper was used to measure the length (mm), width (mm), and height (mm) of each duck crested cushion and to calculate the volume (mm^3^). Crested-cushion length was measured as the distance between the anterior and posterior sagittal ends of the cushion tissue, cushion width was measured as the distance from the left to the right end of the coronal plane of the crested cushion tissue, and height was measured as the distance between the lower ends of the coronal plane of the crested-cushion tissue. The volume was recorded as the product of the transverse diameter, longitudinal diameter, and height. A tape measure was used to determine the distance from the top edge of the bill to the front edge of the bottom of the crested cushion as the distance from the mouth, and the distance was determined from the front edge of the bottom of the crested cushion to the left eye as the distance from the left eye. The distance from the right eye and the distance from the left and right ears were determined using the same measurement technique and tools. 

### 2.4. Blood Parameter Measurements

For serum studies, after measuring the position of the crested cushion and body weight of the two groups, approximately 5 mL of blood was collected from the brachial vein of the ducks from two groups via venipuncture. The blood samples were shaken evenly in coagulation-promoting tubes using a refrigerated centrifuge at 3000× *g* for 10 min at 4 °C, and the supernatant serum was decanted and stored at −20 °C. Total cholesterol (TC) and triglyceride (TG) levels in serum were measured by a CHOD-PAP commercial kit. High-density lipoprotein (HDL), low-density lipoprotein (LDL), glycosylated low-density lipoprotein (G-LDL), and very-low-density lipoprotein (vLDL) were detected by the clearance method. Immunoglobulin A (IgA), immunoglobulin G (IgG), immunoglobulin M (IgM), and immunoglobulin D (IgD) were detected by immunoturbidimetric enzyme-linked immunosorbent assay kits. Potassium (K) and sodium (Na) were detected by enzyme colorimetry commercial kits. Chlorine (Cl) was detected by Chloride Test kit (Colorimetric). Phosphorus (P) was detected by a molybdenum blue colorimetric method Phosphorus Kit. The levels of zinc (Zn), selenium (Se), copper (Cu), and calcium (Ca) were determined by colorimetry. Magnesium (Mg) was detected by a Calmagit Magnesium Kit. Iron (Fe) was detected using the Fe Test Kit FerroZine Colorimetric. All kits were from Beijing Huaying Institute of Biotechnology, and the kit catalog numbers are as follows: TC (HY-N0029), TG (HY-N0030), HDL (HY-N0031), LDL (HY-N0032), vLDL (HY-N0033), G-LDL (HY-N0034), IgA (HY-N0048), IgM (HY-N0049), IgG (HY-N0050), IgD (HY-N0051), K (HY-21007), Na (HY-21006), Cl (HY-N0020), P (HY-21008), Zn (HY-21005), Se (HY-21009), Cu (HY-21004), Ca (HY-21001), Mg (HY-21002), and Fe (HY-21003). All indicators were measured using an automatic biochemical analyzer (Mindray BS-420, Shenzhen Mindray Bio-Medical Electronics Co., Ltd., Shenzhen, Guangdong, China). For samples left over from separated serum, we measured the serum SCFA levels and selected propionic acid, isobutyric acid, acetic acid, butyric acid, isovaleric acid, valeric acid, and caproic acid for concentration determination. Samples were analyzed using a gas chromatograph (Thermo Trace 1310, Thermo Fisher Scientific, Waltham, MA, USA) and a mass spectrometer (Thermo ISQ LT, Thermo Fisher Scientific, USA), and the results were handed over to Suzhou Panomix Bio-pharmaceutical Technology Co., Ltd., Suzhou, Jiangsu, China. All the measurements were performed in triplicate.

### 2.5. Statistical Analysis

Statistical analysis was performed using SPSS (version 22.0, SPSS, Inc., Chicago, IL, USA) for experimental data, and one-way analysis of variance was used to analyze the differences between samples. Duncan’s multiple comparison test was used to evaluate the significance of differences between the two groups. All results are expressed as the mean ± standard deviation (SD). The correlation between serum biochemical indices and the number of crested cushions in ducks was examined using the Pearson’s two-sided test. Pearson’s correlation coefficient is denoted by “r”, where r > 0 and r < 0 represent positive and negative correlation, respectively. Further, 0 ≤ |r| < 0.2, 0.2 ≤ |r| < 0.4, 0.4 ≤ |r| < 0.6, 0.6 ≤ |r| < 0.8, and 0.8 ≤ |r| ≤ 1.0 indicate no correlation, weak correlation, moderate correlation, strong correlation, and extremely strong correlation, respectively. Data were considered statistically significant when *p* < 0.05. 

## 3. Results

To determine whether there are significant differences in various indicators between groups with different numbers of crested cushions and whether the number of crested cushions is associated with various indicators, we measured the size and location of the crested cushion of the ducks, and a serum sample was taken to assess biochemical parameters, including the determination of some immune parameters.

### 3.1. Characteristics of Crested Cushion

Egg hatching is an important process in poultry production. Therefore, embryo mortality during incubation can directly affect the economic benefits of duck farms. There were two peaks of mortality during egg incubation. The first death peak occurs from day 2 to 4 of the egg incubation process, which mainly depends on the fertilization and health status of the egg and appropriate environmental conditions, individual embryos are weak or presence of dead eggs from laying; improper egg storage also decreases embryo viability. The second death peak occurs from day 19 to 21 [19,20]. The mortality rate in the second period accounted for approximately half of the total mortality rate during the entire incubation period. During this period, the physiological changes in embryos are severe, and the transition of embryos from allantoic chorionic respiration to pulmonary respiration leads to a dramatic increase in oxygen demand. Many factors lead to the death of embryos, such as temperature, even heating, humidity, and oxygen [21,22]. However, it is unclear whether special traits affect the hatching process. 

Of the 322 duck eggs collected, 36 unfertilized eggs were removed after candling, and 235 ducks were finally hatched, including 141 ducks (73 males and 68 females) with a single crested cushion and 94 ducks (58 males and 36 females) with multiple crested cushions. There were 51 unhatched duck eggs, including 26 ducks with one crested cushion and 21 ducks with multiple crested cushions, which could not be identified because of accidental death. The results are presented in Table 2; the fertilization rate of crested cushion duck eggs was 88.8% and the hatching rate was 82.17%, with a mortality rate of 18.44% in single-crested-cushion ducks and 22.34% in multiple-crested-cushion ducks. 

After 100 days, the weights of all hatched ducklings and the crested cushions were measured. The results are presented in Table 3, which shows that the average weight of the group with multiple crested cushions was higher than that of the group with a single crested cushion (*p* < 0.01). The differences between the size of the crested cushions and the location are shown in Table 3. The larger the number of crested cushions, the larger their size (*p* < 0.01). There was also a significant difference in the distance between the crested cushion and mouth (*p* < 0.05), and the greater the number of crested cushions, the closer was the distance to the mouth. There was no significant difference in the distance between the crested cushion and the left or right eye and between the crested cushion and the left or right ear (*p* > 0.05). 

### 3.2. Comparison of Serum Biochemical Indicators

Ca, P, Na, K, and other mineral elements, as well as Cl and other macroelements, are involved in the normal growth and metabolism of livestock and poultry. Trace elements in the body of animals are low but also play an essential role as they participate in various enzymatic reactions and promote metabolism. We tested the collected serum samples for various biochemical indicators and checked that all indicators were within the specified content range, which represented the reference values for our results. The results showed that the contents of Ca, Zn, and Se in the serum of multiple-crested-cushion ducks were significantly higher than those of single-crested-cushion ducks (*p* < 0.05). It has been shown that Se and Zn are essential nutrients that play an important role in oxidative stress and immunity in addition to having anti-apoptotic activity, thereby enhancing immunity in poultry [23,24,25,26,27]. An excess of Ca can interfere with P absorption [28]. The contents of Mg, Fe, and Cu were significantly higher than those in single-crested-cushion ducks (*p* < 0.01), and there were no significant differences in Na, Cl, and P content between the two groups of ducks (*p* > 0.05) (Table 4). Fe is involved in the synthesis of collagen and acts as a component of a variety of enzymes in animals [29,30]. Cu plays a crucial role in the normal growth, bone development, and metabolism of animals [31,32]. Mg and Cu function similarly, but Mg has a positive effect on the meat quality and body weight of poultry at the developmental stage [33]. The above results showed that the mineral element content in the serum of the multiple-crested-cushion group, and the resulting effects were mostly better than those of the single-crested-cushion group. In addition, the correlation analysis between the number of crested cushions and trace elements showed that the number of crested cushions was significantly positively correlated with the contents of Ca, Zn, Se, and Mg (*p* < 0.05) and highly significantly positively correlated with Fe and Cu content (*p* < 0.01). (Table 5). 

The results of basic serum biochemical parameters showed that the contents of TC and TG in multiple-crested-cushion ducks were higher than those in single-crested-cushion ducks, but only the TG content was significantly different between the two groups (*p* < 0.05); moreover, there was no significant difference in TC (*p* > 0.05). As shown in Table 6, there was no significant difference in HDL, LDL, and vLDL content (*p* > 0.05), but the contents in multiple-crested-cushion groups were slightly higher than those in single-crested-cushion groups; results of the difference analysis showed that G-LDL content was significantly higher in the serum of single-crested-cushion ducks than in that of multiple-crested-cushion ducks (*p* < 0.01). In the background of the above results, we continued to correlate the data shown in Table 7 and found that the number of crested cushions was significantly positively correlated with TG content in serum (*p* < 0.05) and significantly negatively correlated with G-LDL (*p* < 0.01). 

To determine immune function and resistance to infection, we measured levels of serum immune markers, including immunoglobulin A (IgA), immunoglobulin G (IgG), immunoglobulin M (IgM), and immunoglobulin D (IgD). The effect of the number of crested cushions on the serum immune parameters is shown in Table 6. The four immune parameters were significantly higher in the multiple-crested-cushion group than in the single-crested-cushion group, and the difference was extremely significant (*p* < 0.01). A correlation analysis was subsequently performed, and the results were the same as those of previous results (Table 7). Thus, IgA, IgG, IgM, and IgD levels were significantly positively correlated with the number of crested cushions (*p* < 0.01). The higher the number of crested cushions, the higher is the immunoglobulin content in the serum. In addition, lymphocytes in the innate immune system of birds can produce a variety of antibodies, such as IgM, IgG, and IgA, as the first line of defense against inflammation [34]. 

Current studies on SCFAs in poultry have focused on gut flora and gut health, which directly affect the growth of poultry. The aim of this study is to screen out RWC ducks which are better in all aspects, so the content of SCFAs in duck serum is also an important criterion for evaluating duck health [35,36]. Next, we measured the levels of the remaining serum SCFAs (Table 8). Propionic acid, isobutyric acid, acetic acid, butyric acid, isovaleric acid, valeric acid, and caproic acid were selected for concentration determination, but the concentrations of isovaleric acid, valeric acid, and caproic acid in the detection results were lower than the lowest quantitative limit; thus, the final serum short-chain fatty acid content results were only reported in terms of propionic acid, isobutyric acid, acetic acid, and butyric acid content. The results showed that the propionic acid and acetic acid content in the serum of the multiple-crested-cushion group was significantly higher than that in the single-crested-cushion group (*p* < 0.05), and the butyric acid content was significantly higher than that in the single-crested-cushion group (*p* < 0.01), and there was no significant difference in isobutyric acid content between the two groups (*p* > 0.05). 

## 4. Discussion

The crested cushion is a visible trait that is present in many birds and is characterized by a cluster of villous projections on the head of birds, forming a hair bulb; some birds have parenchymal tissue in the cluster of villous projections [37,38,39]. As a precious germplasm resource, the RWC duck has strong ornamental value. However, while examining crested-cushion duck crest traits, we found that ducks have high mortality and movement disorders. Our results showed that duck embryos with multiple crest cushions had a higher mortality than those with single crest cushions. Lancaster [40] found that homozygous offspring of the crested cushioned breed of domestic duck (*Anus platyrhynchos f. doni.*) had encephalocele, disability, or loss of the upper beak, resulting in an inability to successfully incubate, with high mortality and central nervous system defects. Therefore, we speculate that the number of crest cushions may have an effect on the duck egg mortality rate, and individuals with a large number of crest cushions are more likely to die during the embryonic period. This result is likely due to cranial vault defects caused by the compression of brain tissue during embryonic growth, which subsequently blocks brain development. 

When measurements of the crest cushion, with respect to number, size, and position, were correlated with morphological traits, we found that ducks with multiple crest cushions had an extremely significantly higher body weight. Another highly significant difference was the distance of the crest cushions from the mouth, which was closer in ducks with multiple crested cushions. Bartels et al. [41] found that the deposition rate of intracranial tissue in domestic ducks with crested-cushion traits was extremely high, but there was no significant correlation between the crested-cushion volume and intracranial tissue deposition (*p* > 0.05). These results preliminarily confirmed that the number of crested cushions had a negative impact on the hatching and survival rate of ducks. However, duck with multiple crest cushions are more advantageous during growth. To explore this effect, we measured serum biochemical parameters in ducks with different numbers of crested cushions. Studies by Puvača and Stana. [27] have shown that minerals are important for optimizing the performance of livestock and poultry. The initial symptoms of mineral imbalance and malnutrition are growth inhibition, slow weight gain, inefficient feed use, and reproductive efficiency. Balan and Sik-Han [42] found that zinc deficiency was associated with growth retardation, hypogonadism, and immune dysfunction. Appropriate addition of trace elements, especially selenium, to feed can cause physiological changes in the muscle tissue and improve meat quality in poultry. It is worth noting that K, in addition to regulating osmolality, maintains excitability of nerves and muscles, and the main function of Se is antioxidation, toxicity reduction of some metals and tumors, and immunity enhancement [43,44]. Our results show that the contents of minerals, such as Mg, Fe, and Cu, and to a lesser degree, Ca, Zn, and Se, were higher in ducks with multiple crest cushions. Thus, we can tentatively deduce that RWC ducks with multiple crest cushions have superior health, reproductive efficiency, growth, and immune function. We speculate that the effect of the number of crest cushions on the indicators above may be only one aspect, and the results may be due to a synergy with other factors, which remains to be further investigated. Ducks with multiple crest cushions were found to have significantly higher TG content in their blood serum. Qiao et al. [45] found that TG content in plasma was significantly correlated with body fat mass, and excessive body fat deposition resulted in weight gain; therefore, we speculated that this was why the mean body weight of the multiple-crested-cushion group was higher than that of the single-crested-cushion group. Unlike our results, Al-Ani et al. [46] found that G-LDL was significantly associated with HDL, TG, and LDL, whereas our correlation analysis showed that several other indicators, except G-LDL, were significantly associated, suggesting that the reason for the different results may be due to the different species studied. It is worth noting that thus far, research on G-LDL has focused on human diabetes, atherosclerosis, and other diseases, and no study has been conducted on livestock and poultry.

The crested cushion is a tumor-like tissue. Schilling et al. [47] found that at the final stage of hatching, embryos have immunity and can mount innate and adaptive immune responses to pathogens. Fu et al. [48] found that the crested ibis (*Nipponia nippon*), an endangered bird worldwide, has a very high embryonic and nestling mortality rate, and comparing living and dead samples using high-throughput sequencing revealed that genes for glycan biosynthesis and metabolism (HYAL1 and HYAL4), as well as the immune system (JAM2), are associated with embryonic and nestling development. Serum immunoglobulins, especially IgA, IgG, and IgM produced by B cells, are important parameters reflecting the humoral immune status of animals and are associated with their important role in immune function and resistance to infection [49,50]. Immler et al. [51] found that metabolism in dairy cows was associated with the passive transfer of immunoglobulins, and newborn calves fed more than 2 L colostrum showed higher serum immunoglobulin levels, suggesting that immunoglobulin content in serum can indirectly indicate the health of animals. One animal study demonstrated that saturated fatty acids in visceral adipose tissue (VAT) increased IgM levels via stimulation of the B-cell Toll-like receptor 4 (TLR-4) in a manner similar to lipopolysaccharide. This finding supports the hypothesis that increased TG levels leads to increased IgM concentration [52]. Elevated serum TG levels promote oxidative stress, which may lead to increased IgM concentrations, indirectly indicating a positive correlation between TC and immunoglobulin content in serum, which is consistent with our findings [53]. The result of our analysis indicates that multiple crested cushions corresponded with the higher immunoglobulin content in the serum. Lymphocytes in the innate immune system of birds can produce a variety of antibodies, such as IgM, IgG, and IgA, as the first line of defense against infection and inflammation. It has been found in human nephropathy models that IgG levels are positively correlated with serum albumin and serum IgA levels and negatively correlated with serum cholesterol levels in patients with immunoglobulin A nephropathy (IgAN) [54]. This was not noted in our findings, which tentatively indicates that our experimental results are credible and play a positive regulatory role in the health and immunity of ducks with an increase in the number of crested cushions. SCFAs are mainly composed of acetate, propionate, and butyrate and are beneficial for body weight control and lipid homeostasis as well as playing an important role in poultry gut health [55,56,57]. The inflammatory response is reduced by inducing changes in the gut microbiota of chickens and indirectly mediating the gut microbiota biological barrier by promoting the secretion of mucins and antimicrobial peptides in the gut [35,58]. Studies in humans and animals have found that propionic acid is used as a major substrate for de novo fat synthesis [59,60]. Studies have also shown that SCFAs, particularly butyric acid, can improve insulin resistance in mice fed a high-fat diet, possibly by maintaining mitochondrial activity and energy expenditure [61]. The contents of propionic acid, acetic acid, and butyric acid in the serum from the experimental ducks were higher with a higher number of crest cushions. These results further indicate that the content of short-chain fatty acids plays a positive role in regulating the health of animals, and combined with our results, it also indirectly shows that the greater the number of crested cushions, the more advantageous the impact on RWC crested-cushion duck health. 

Based on our previous work and the aforementioned results in the present study, we considered that a greater number of crests leads to brain exposure, which causes the body’s protective mechanism to compensate for the purpose of protecting the brain from viruses or bacteria, and leads to the increase in some key minerals and immune indicators related to the immune status through adaptive evolution. Another idea is that whenever the RWC duck grows a crested cushion, it stimulates the body’s immune process again. Therefore, the number of crested cushions can have an impact on mineral metabolism and immune status in RWC ducks.

## 5. Conclusions

Our results show that the number of crested cushions could be used to directly reflect the size of the crested cushions. With an increase in the number of crested cushions, there is increased compression of the skull and brain by the crested cushion during the embryonic period, leading to a higher mortality. However, ducklings with multiple crested cushions gained weight more quickly after hatching. In addition, the number of crested cushions had a positive effect on various serum parameters, such as various mineral elements, as well as triglyceride and immunoglobulin levels. Therefore, it is preliminarily speculated that the number of crested cushions may indirectly affect the health of Runzhou crested white ducks and the advantages and disadvantages of their immune performance. This study lays a foundation to further improve the quality of RWC duck breeding. In the future, we may focus on breeding and expanding the population of RWC ducks according to the number of crested cushions to breed RWC ducks with better performance in all aspects.

## Figures and Tables

**Table 1 animals-13-00466-t001:** Table of nutritional value of feed in each stage.

Items	0–42 d	43–100 d
	Ingredient (%)	
Corn	10.32	34.23
Wheat middling	15.41	13.45
Wheat bran	—	25
Rice noodles	35.21	10
Rice bran	15.81	4
Peanut meal	—	2.69
Corn gluten meal	—	5
Soybean meal	12.63	4.22
Nucleotide slag	2	—
Limestone powder	1.52	1.9
Calcium hydrogen phosphate	1.1	1.01
Compound premix	6	6
	Nutritional level (%)	
Crude protein	185	170
Crude fat	20	35
Crude fiber	60	70
Crude ash	90	100
Calcium	10	10
Phosphorus	5	4.5
Sodium chloride	6	6
Methionine	4	2.8
Moisture	140	130

“d” represents the “day” in the first row.

**Table 2 animals-13-00466-t002:** Effect of crested cushion number on reproductive performance of RWC ducks.

Items	Incubation Number	Fertilization Number	Fertilization Rate (%)	Number of Incubations	Hatching Rate (%)	Number of Deaths	Mortality Rate (%)
Single crest	/	/	/	141	49.30%	26	18.44%
Multiple crest	/	/	/	94	32.87%	21	22.34%
Total	322	286	88.8%	235	82.17%	51	21.70%

**Table 3 animals-13-00466-t003:** Effect of the number of crested cushions on the weight of ducks and of crested cushion location.

Items	Single Crest	Multiple Crest	*p*-Value
Sample Size (n)	127	79	
Body weight (g)	1484.72 ± 128.17 ^b^	1629.47 ± 76.77 ^a^	<0.01
Volume of crest cushion (cm^3^)	3.10 ± 0.42 ^B^	16.51 ± 1.65 ^A^	<0.01
Distance from left eye to crest cushion (cm)	2.98 ± 0.32	2.88 ± 0.43	0.213
Distance from right eye to crest cushion (cm)	3.03 ± 0.40	2.88 ± 0.38	0.07
Distance from left ear to crest cushion (cm)	3.53 ± 0.43	3.59 ± 0.43	0.526
Distance from right ear to crest cushion (cm)	3.40 ± 0.46	3.55 ± 0.50	0.126
Distance from mouth to crest cushion (cm)	6.61 ± 0.71 ^a^	6.26 ± 0.73 ^b^	<0.05

^a, b, A, B^ Within a row, for each factor, different superscript letters represent significant differences, with capital letters representing extremely significant differences (*p* < 0.01) and small letters representing significant differences (*p* < 0.05). All results presented as mean ± SD except sample size. SD: standard deviation of sample.

**Table 4 animals-13-00466-t004:** Effect of crested-cushion number on trace elements in serum biochemical indexes.

Items	Single Crest	Multiple Crest	*p*-Value
Na (mmol/L)	133.38 ± 19.70	132.46 ± 20.49	0.835
Cl (mmol/L)	85.68 ± 8.50	86.22 ± 7.53	0.748
P(mmol/L)	0.80 ± 0.05	0.88 ± 0.06	0.282
Ca (mmol/L)	2.19 ± 0.16 ^b^	2.89 ± 0.29 ^a^	<0.05
Mg (mmol/L)	0.62 ± 0.03 ^B^	0.89 ± 0.09 ^A^	<0.01
Fe (umol/L)	14.81 ± 1.04 ^B^	22.84 ± 2.04 ^A^	<0.01
Cu (umol/L)	1.62 ± 0.14 ^B^	3.07 ± 0.30 ^A^	<0.01
Zn (umol/L)	17.69 ± 5.11 ^b^	20.38 ± 6.95 ^a^	<0.05
Se (ug/L)	14.57 ± 1.07 ^b^	18.42 ± 1.66 ^a^	<0.05

^a, b, A, B^ Within a row, for each factor, different superscript letters represent significant differences, with capital letters representing extremely significant differences (*p* < 0.01) and small letters representing significant differences (*p* < 0.05). All results presented as mean ± SD except sample size. SD: standard deviation of sample.

**Table 5 animals-13-00466-t005:** Correlation analysis of crested-cushion number with macroelements, trace elements, and mineral elements.

	K	Na	Cl	P	Ca	Mg	Fe	Cu	Zn	Se	Number of Crest
K	1										
Na	0.261 *	1									
Cl	0.443 **	0.928 **	1								
P	0.367 **	0.551 **	0.663 **	1							
Ca	0.173	0.537 **	0.637 **	0.681 **	1						
Mg	0.159	0.570 **	0.636 **	0.682 **	0.967 **	1					
Fe	−0.026	0.352 **	0.426 **	0.165	0.699 **	0.676 **	1				
Cu	0.128	0.458 **	0.549 **	0.370 **	0.649 **	0.624 **	0.559 **	1			
Zn	0.173	0.589 **	0.719 **	0.498 **	0.786 **	0.774 **	0.771 **	0.712 **	1		
Se	−0.005	0.366 **	0.375 **	0.182	0.620 **	0.535 **	0.586 **	0.834 **	0.612 **	1	
Number of crest	0.017	−0.023	0.034	0.117	0.229 *	0.315 **	0.392 **	0.529 **	0.219 *	0.233 *	1

In each row, for the superscript of a number, “*” represents a significant correlation between the two, “**” represents a highly significant correlation between the two, and the “−“ before the number represents a negative correlation between the two; no sign in front of the number represents a positive correlation between the two.

**Table 6 animals-13-00466-t006:** Effect of crested cushion on lipoprotein and immunoglobulin in serum biochemical parameters.

Items	Single Crest	Multiple Crests	*p*-Value
TC (mmol/L)	3.27 ± 0.13	3.51 ± 0.18	0.261
TG (mmol/L)	1.26 ± 0.09 ^b^	1.71 ± 0.16 ^a^	<0.05
HDL (mmol/L)	1.83 ± 0.09	1.85 ± 0.11	0.878
LDL (mmol/L)	1.00 ± 0.05	1.17 ± 0.08	0.071
vLDL (mmol/L)	0.46 ± 0.06	0.47 ± 0.08	0.271
G-LDL(Index)	29.61 ± 3.98 ^A^	27.14 ± 1.14 ^B^	<0.01
IgA (g/L)	1.23 ± 0.05 ^B^	1.78 ± 0.10 ^A^	<0.01
IgG (g/L)	3.68 ± 0.11 ^B^	4.55 ± 0.16 ^A^	<0.01
IgM (g/L)	0.76 ± 0.04 ^B^	1.00 ± 0.07 ^A^	<0.01
IgD(mg/L)	31.50 ± 3.40 ^B^	35.36 ± 4.85 ^A^	<0.01

^a, b, A, B^ Within a row, for each factor, different superscript letters represent significant differences, with capital letters representing extremely significant differences (*p* < 0.01) and small letters representing significant differences (*p* < 0.05). All results presented as mean ± SD except sample size. SD: standard deviation of sample.

**Table 7 animals-13-00466-t007:** Correlation analysis of crested-cushion number with lipoprotein and immunoglobulin in serum biochemical parameters.

	TC	TG	HDL	LDL	vLDL	GLDL	IgA	IgG	IgM	IgD	Number of Crest
TC	1										
TG	0.229	1									
HDL	0.847 **	0.240 *	1								
LDL	0.604 **	0.518 **	0.406 **	1							
vLDL	0.581 **	0.466 **	0.350 **	0.954 **	1						
GLDL	−0.102	−0.161	0.030	−0.148	−0.099	1					
IgA	0.008	0.469 **	0.084	0.431 **	0.341 **	−0.258 *	1				
IgG	0.075	0.460 **	−0.008	0.611 **	0.515 **	−0.460 **	0.646 **	1			
IgM	−0.133	0.582 **	−0.094	0.299 **	0.231 *	−0.229 *	0.744 **	0.640 **	1		
IgD	0.003	0.534 **	−0.036	0.548 **	0.470 **	−0.326 **	0.866 **	0.925 **	0.834 **	1	
Number of crest	0.124	0.287 *	0.017	0.188	0.115	−0.292 **	0.492 **	0.445 **	0.302 **	0.425 **	1

In each row, for the superscript of a number, “*” represents a significant correlation between the two, “**” represents a highly significant correlation between the two, and the “−“ before the number represents a negative correlation between the two; no sign in front of the number represents a positive correlation between the two.

**Table 8 animals-13-00466-t008:** Effect of crown cushions on short-chain fatty acids in serum biochemical parameters.

Items	Single Crest	Multiple Crest	*p*-Value
Propionic acid	0.099 ± 0.011 ^b^	0.143 ± 0.016 ^a^	<0.05
Isobutyric acid	0.055 ± 0.009	0.035 ± 0.003	0.130
Acetic acid	4.021 ± 0.258 ^b^	5.026 ± 0.391 ^a^	<0.05
Butyric Acid	0.046 ± 0.003 ^B^	0.068 ± 0.003 ^A^	<0.01

^a, b, A, B^ Within a row, for each factor, different superscript letters represent significant differences, with capital letters representing extremely significant differences (*p* < 0.01) and small letters representing significant differences (*p* < 0.05). All results presented as mean ± SD except sample size. SD: standard deviation of sample.

## Data Availability

All available data are incorporated in the manuscript.

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
