# Peer review of "Effects of the Number of Crested Cushions in Runzhou White-Crested Ducks on Serum Biochemical Parameters"

_animals, 2023, doi:10.3390/ani13030466_

Round 1
Reviewer 1 Report
1. In “2.3. Crested Cushion Data Collection” part, the eggs that did not successfully hatch were collected separately and broken, and the number of embryos inside and the number of crested cushions were counted, did the author have ruled out other reasons that may lead to failure of incubation, such as pathogens?
2. Some small font and language errors need to be rechecked.
Reviewer 2 Report
In this work, Lou et al showed that the number of crested cushions in Runzhou ducks affects weight and mortality. Similarly, the authors that the levels of immunoglobulin and triglycerides were higher in ducks with higher crested cushions. Although the work presents novel findings as regards the Runzhou ducks, I do have some concerns about the findings.
Minor issues
1-Why was day 100 chosen for the testing?
2-There should be a ref on lines 51-52, lines 54-55, and lines 74-76
3-Se in table 4 should be formatted
4-The statistical tool used should be written
Major concerns
1-In section 2.3, the authors showed that there was higher mortality in unhatched ducks with a higher number of crests without providing sufficient information on the mother duck. There should be information on the number of crest of the mother duck from which the eggs were obtained.
2-Most of the parameters were measured only in the single crest and multiple crests. However, I think non-crested ducks would serve as a better baseline.
3-Why do the immunoglobulin levels increase alongside triglycerides? I think measuring at different time points would be a better representation than just a single time point. Secondly, does the crest have any roles in immunology or lipid synthesis? The authors need to elaborate on this.
4-The basis for testing SCFAs should be specified
Reviewer 3 Report
The article is of scientific and practical interest. It is necessary to clarify the research methods, discussion and conclusion in accordance with the comments in the attached file. After correcting the comments and additions, the article can be published.

Round 2
Reviewer 2 Report
The authors have sufficiently addressed my concerns.